# Application of a Chitosan–Cinnamon Essential Oil Composite Coating in Inhibiting Postharvest Apple Diseases

**DOI:** 10.3390/foods12183518

**Published:** 2023-09-21

**Authors:** Wanli Zhang, Gulden Goksen, Yuanping Zhou, Jun Yang, Mohammad Rizwan Khan, Naushad Ahmad, Tao Fei

**Affiliations:** 1School of Food Science and Engineering, Hainan University, Haikou 570228, China; 2College of Food Science and Nutritional Engineering, China Agricultural University, Beijing 100083, China; 3Department of Food Technology, Vocational School of Technical Sciences at Mersin Tarsus Organized Industrial Zone, Tarsus University, 33100 Mersin, Turkey; 4Department of Chemistry, College of Science, King Saud University, Riyadh 11451, Saudi Arabia

**Keywords:** apple fruit, chitosan coating, cinnamon essential oil, postharvest

## Abstract

The purpose of this study was to explore the film-forming properties of cinnamon essential oil (CEO) and chitosan (CS) and the effect of their composite coating on postharvest apple diseases. The results demonstrated that the composite coating exhibits favorable film-forming properties at CEO concentrations below 4% (*v*/*v*). The effectiveness of the composite coating in disease control can be attributed to two factors: the direct inhibitory activity of CEO against pathogens in vitro and the induced resistance triggered by CS on the fruits. Importantly, the incorporation of CEO did not interfere with the induction of resistance by CS in harvested apples. However, it is noteworthy that the inhibitory effect of the CS–CEO composite coating on apple diseases diminished over time. Therefore, a key aspect of enhancing the preservation ability of fruits is improving the controlled release properties of CEO within CS coatings. This will enable a sustained and prolonged antimicrobial effect, thereby bolstering the fruit preservation capabilities of the composite coatings.

## 1. Introduction

Fruits continue to exhibit living biological activities following harvest. As they undergo the process of maturation and aging, their natural defenses against diseases gradually diminish, making them susceptible to infections caused by pathogenic fungi. Consequently, this leads to the occurrence of fruit rot and economic losses of substantial magnitude [1,2,3]. Apples have good flavor and high nutritional value and are loved by consumers around the world. However, apple fruits are susceptible to fungal infection during postharvest transportation and storage stages, leading to large-scale decay. Fungi, including *Penicillium expansum*, *Penicillium digitatum*, *Alternaria alternata*, and *Botrytis cinerea*, are the primary causal agents responsible for the postharvest decay of various fruits, such as peaches, apricots, apples, and tomatoes [4]. At present, the control of postharvest fruit diseases primarily relies on the application of certain synthetic chemical fungicides. However, the acceptance of fruits treated with these chemical agents remains relatively low among consumers due to concerns regarding their safety [5]. As a result, the focus in the control of fruit diseases after harvesting has transitioned toward the creation and execution of natural active agents and physicochemical techniques that possess antimicrobial properties. Examples of such approaches include the use of plant essential oils [6], exposure to UV radiation [7], the deployment of lysozyme and peroxidase [8], treatment with natural inducers [2], and the use of natural metal nanoparticles [9], among others.

Chitosan (CS) coatings are used as a technique for controlling postharvest fruit diseases. Nevertheless, the effectiveness of chitosan in directly suppressing fungi is relatively weak. The primary aspect of CS’s contribution to the control of postharvest fruit diseases is its ability to induce fruit resistance [10]. Natural plant resistance is induced by CS, a component of fungal cell walls. It decreases disease incidence by promoting disease resistance enhancement inside the fruits themselves. Plant defense mechanisms rely on crucial enzymes, including chitinase, β-l,3 glucanase, and phenylalanine ammonia lyase (PAL), as well as the synthesis of polyphenols and lignin through phenylalanine metabolism. These components are essential in providing resistance against diseases in plant tissues [11]. Previous studies have demonstrated that the application of 1% CS combined with spermine coating treatment yielded the most promising outcomes in terms of controlling postharvest anthracnose in mango fruit. Additionally, composite coating treatment significantly reduced spot diameter and increased the total phenolic content of mango fruit during storage. The triggering of peroxidase (POD) and phenylalanine ammonia lyase (PAL) activities, which are involved in phenylpropane metabolism, led to an improvement in the resistance of apple fruit against pathogenic bacteria [12].

Although the application of a CS coating provides certain benefits by enhancing the resistance of postharvest fruits against diseases, its ability to manage postharvest fruit diseases is significantly restricted owing to its insufficient antifungal efficacy. In contrast, a promising and effective strategy for postharvest fruit disease control is the application of a composite coating consisting of CS and a plant essential oil. This is essentially because the antifungal activity of the composite coating is greatly enhanced by the incorporation of a plant essential oil. Moreover, CS serves as a carrier for the plant essential oil, thereby mitigating any negative impact it may have on fruit quality [13]. Both the direct antifungal properties of the plant essential oil and the induced resistance effect of CS are key to the success of CS–plant essential oil composite coatings in preventing postharvest fruit diseases [14]. It is noteworthy that the volatile molecules found in plant essential oils might pass through the composite coating and permeate the fruit. The fruit’s defense mechanism may, thus, be affected. While the direct antifungal activities of these essential oils have been the main focus of previous research, studies have mostly ignored the potential of these essential oils to induce resistance in postharvest fruits. Furthermore, a limited number of studies have investigated the kinetics of CS coatings that incorporate plant essential oils in preventing postharvest fruit diseases. This study aims to examine the impact of a CS–cinnamon essential oil (CEO) composite coating on the control of postharvest fungal diseases in apples. It evaluates the in vitro direct antifungal impact of the CS–CEO composite covering as well as the in vivo induced resistance in order to comprehend the complex mechanisms underlying its efficacy. Therefore, this work innovatively verifies the influencing factors of a CS–CEO coating in postharvest apple diseases through in vivo and in vitro experiments.

## 2. Materials and Methods

### 2.1. Materials

Apple fruits (Huang Yuanshuai) were obtained from the experimental garden located in Pinggu District, Beijing, and they served as the chosen specimens for this series of experiments. After harvesting, the fruits were promptly transported to the laboratory to ensure their freshness and integrity. The fruits were carefully selected for their average size, color, and maturity level, as well as their lack of visible mechanical injuries and evidence of pests and diseases. The fruits were surface-disinfected by soaking them in 70% (*v*/*v*) ethanol for 2 min before experiments. Following this, they were thoroughly washed with tap water and left to dry naturally at room temperature. Cinnamon essential oil was purchased from Shenzhen Dingcheng Spice Co (cinnamaldehyde content: 88.9%). The *Penicillium expansum*, *Penicillium digitatum*, *Alternaria alternata*, and *Rhizopus stolonifer* used in the experiments were all isolated from related fruits in our laboratory and underwent morphological identification, ensuring their relevance and reliability. All other chemicals and solvents of analytical purity were acquired from Beijing Reagent Co. (Beijing, China).

### 2.2. Film-Forming Properties of CEO–CS Coatings

The CS film-forming solutions incorporated with CEO were prepared according to Zhang et al. [15]. Various concentrations of CEO (1%, 2%, and 4% *v*/*v*) were added to a 2% CS solution. To facilitate emulsification, an emulsifier in the form of 20% Tween 80 (*v*/*v* based on essential oil) was added, and the mixture was subjected to mechanical stirring for 30 min. The particle size and zeta potential were measured using a nanoparticle sizer to better understand the properties of the various CS film-forming solutions.

Gradually, 10 mL of the film-forming solutions was poured into 8-cm-diameter polystyrene plastic petri dishes. The coatings were then dried for 10 h at 35 °C. Next, the coatings were stored at a constant temperature of 25 ± 2 °C and a relative humidity of 60 ± 5% for 48 h to achieving a stable moisture balance. To evaluate the water solubility of the coating samples, they were first subjected to drying until a state of constant weight (*M*_0_) was achieved at a temperature of 110 °C. Next, 1 cm × 1 cm coated strips were submerged in 100 mL of distilled water and gently agitated at 50 rpm/min and 25 °C for 12 h. After soaking, the coatings were removed carefully and dried again until a consistent weight (*M*_1_) was achieved, at 105 °C. The water solubility of the coatings was calculated using the following equation:(1)$$\mathrm{Water}\;\mathrm{content}\;(\%)=\frac{M_{0}-M_{1}}{M_{0}}\times 100$$

The water vapor permeability (WVP) of the coatings was assessed according to Kadam et al. [16] with slightly modifications [16]. The coating samples were hermetically sealed at the mouth of a weighing cup measuring 4 cm in diameter and 4 cm in depth. To the cups, 3 g of CaCl_2_ was added, and the cups containing the coating samples were then placed in a desiccator maintained at 20 °C and 70% relative humidity. Saturated potassium iodide acted as a moisture absorber inside the desiccator. The cups were carefully weighed at regular intervals of 4 h, and this process was repeated over 48 h. The WVP of the coatings was computed using the following equation:(2)$$\mathrm{Water}\;\mathrm{vapor}\;\mathrm{permeability}\;=\frac{M\times x}{t\times A\times \Delta p}$$

M in Equation (2) is the weighing cup weight difference (g), *x* is the coating thickness (m), *A* is the coating permeation area (m^2^), Δ*p* is the permeation area vapor pressure difference (2.339 Pa at 20 °C), and *t* is the permeation time (s).

The mechanical properties of the coatings were analyzed, as described by Shah et al. [17]. The tensile strength and elongation at break of the samples were measured using a physical property analyzer (CT3, Brookfield Ltd., USA), and the samples were then precisely cut into rectangles 60 mm × 8 mm in size. The effective length of the stretching operation was 30 mm, and the stretch rate was held constant at 1.0 mm/s. After that, the tensile strength and elongation at break were calculated using appropriate equations. All tests were carried out five times.
(3)$$\mathrm{Tensile}\;\mathrm{strength}\;(\mathrm{MPa})\;=\frac{F}{x\times w}$$
(4)$$\mathrm{Elongation}\;\mathrm{at}\;\mathrm{break}\;(\%)\;=\frac{\Delta x}{x}\times 100$$

F in Equation (3) is the maximum force (N) when the coating ruptures, *x* is the coating thickness (mm), *w* is the width of the coating sheet (mm), Δ*x* in Equation (4) is the increase in length at the rupture point (mm), and *x* is the initial length of the coating sheet (mm).

### 2.3. In Vitro Antifungal Activity of CS Films with Different CEO Concentrations

*Penicillium expansum*, *Penicillium digitatum*, *Alternaria alternata*, and *Rhizopus stolonifer* were inoculated on sterilized and solidified PDA culture medium at a constant temperature of 28 °C. After culturing for 7 days, distilled water was added to scrape the colonies to prepare a spore suspension with a concentration of 10^5^~10^6^ mL^−1^. A pipette was used to add 100 μL of various spore suspensions to the solidified PDA, and a coating rod was used to spread the spore suspension evenly. CS films containing CEO at different concentrations were prepared by referring to the method in Section 2.2. A sterile puncher was used to make CS–CEO composite films of different concentrations into 0.5-cm-diameter discs, and they were placed in the center of a petri dish. The fungal culture medium was cultured at a constant temperature of 28 °C for 72 h. The diameter of the inhibition zone was measured.

### 2.4. Apple Fruit Coating Treatment and Inoculation

We followed a previously described procedure for applying coatings to apples after harvest [15]. For apple fruit inoculation, a precise set of instructions was followed. Equidistantly spaced holes, each measuring 2 mm in diameter and 4 mm in depth, were meticulously created by puncture in the middle equatorial region of the fruit using a sterilized needle. Subsequently, each hole was inoculated with 15 μL of *P. expansum* spore suspension at a concentration of 1 × 10^4^ spores mL^−1^ using a microinjector. After inoculation, the fruit was kept in a controlled environment at constant room temperature and 70% relative humidity. Disease development was tracked by measuring the spot diameters and disease indices of the fruit daily. The 18 fruits were sampled every 2 days during storage in each group. Pieces of fruit were carefully cut out that were 2 cm wide around wounds, and then frozen for further analysis.

### 2.5. Extraction of PAL and Determination of PAL Activity

The extraction of phenylalanine ammonia lyase and the determination of the activity of phenylalanine ammonia lyase (PAL) were performed, as described by Zhang et al. [4] with some modifications. For extraction, 3 g of frozen fruit pulp tissue was combined with 5 mL of extraction solution. This solution comprised a boric acid–borax buffer with a concentration of 100 mmol L^−1^ and possessed a pH of 8.8. Additionally, the extraction solution was prepared with 5 mmol L^−1^ of β-mercaptoethanol, 2 mmol L^−1^ of EDTA, and 40 g L^−1^ of polyvinylpyrrolidone. The resulting mixture was then ground and homogenized via low-temperature shaking for 10 min. After that, the mixture underwent centrifugation at 4 °C for 20 min. The supernatant was collected and used as the extract for PAL measurement. To determine PAL enzyme activity, a reaction mixture was prepared by combining 3 mL of 50 mmol L^−1^ of boric acid–borax buffer at pH 8.8, 0.5 mL of 20 mmol L^−1^ of l-phenylalanine, and 0.5 mL of the enzyme extract obtained in the previous step. Absorbance was measured at a wavelength of 290 nm. PAL enzyme activity was obtained from the measured absorbance value. The PAL enzyme activity was expressed as U mg^−1^ pro, signifying the enzyme activity per milligram of protein present in the sample.

### 2.6. Determination of Total Phenolic Content of Apples

The total phenolic content (TPC) was assessed according to Folin–Ciocalteu’s procedure, as previously described by Singleton and Rossi (1965), with slight modifications [18]. Briefly, 0.5 mL of the extract was mixed with 2.5 mL of 1:10 diluted Folin–Ciocalteu’s phenol reagent, 2.0 mL of 7.5% (*w*/*v*) Na_2_CO_3_ was added, and the mixture was kept for 5 min at 50 °C, followed by measuring absorbance at 760 nm. A standard curve of gallic acid was drawn to estimate the phenol content. The TPC results were expressed as milligrams of gallic acid equivalent (GAE) per kg.

### 2.7. Extraction and Determination of GLU and CHI in Apples

Glucanase (GLU) activity was determined, as prescribed by Jiao et al. [19] with some modifications. A total of 6 g of frozen fruit pulp tissue was precisely weighed and subsequently homogenously ground. To initiate the extraction process, cooled extract (3 mL) comprising sodium acetate buffer (100 mmol/L, pH 5.2) containing EDTA (1 mmol/L), β-mercaptoethanol (5 mmol/L), and ascorbic acid (1 g/L) was added. Following centrifugation at 12,000× *g* for 20 min at 4 °C, the resulting supernatant was collected. Subsequently, dialysis treatment was performed by transferring the supernatant to a dialysis bag (MWCO 142, 8000–14,000, Serva dialysis tubing) and immersing it in distilled water at 4 °C for 24 h. This dialysis process aimed to eliminate salt ions present in the sample. GLU activity was determined based on the enzyme-catalyzed hydrolysis of kombucha polysaccharides, with the production of 10^−9^ mol of glucose equivalents per second being defined as one GLU activity unit (U). The GLU activity was expressed as U per gram of protein (U g^−1^ pro). Similarly, chitinase (CHI) crude enzyme solution was extracted using a similar methodology and referred to the method described by Ma et al. [20]. CHI activity was detected by assessing the production of 10^−9^ mol of N-acetylamino glucosamine equivalents per second resulting from the decomposition of chitin. This measure was defined as one CHI activity unit (U), and the CHI activity was expressed as U per gram of protein (U g^−1^ pro).

### 2.8. Antimicrobial Kinetics of CS–CEO Coatings

Apples with CS coatings containing 4% CEO were perforated and placed at room temperature under ventilated conditions, and the change in antimicrobial ability was measured every 3 days using the inhibition circle method. The antimicrobial ability of the composite coatings was compared by inoculating the coated apples after storage at room temperature and a humidity of about 70% for 5 and 10 days and after direct coating for 0 days.

### 2.9. Statistical Analysis

The experiments included performing a triplicate test on all samples. Each sample was subjected to analysis in at least three parallel groups. The resulting data were then represented as the mean value along with its corresponding standard deviation. The graphing software used in this study was OriginPro 2018. The statistical software SPSS 18.0 was used to conduct a one-way analysis of variance (ANOVA). Duncan’s method was used to assess significant distinctions among several groups of samples, with a significance level of *p* < 0.05.

## 3. Results and Discussion

### 3.1. Particle Size of the CS–CEO Composite

The size of particles plays a vital role in indicating the level of uniformity and stability within an emulsion system. Figure 1A demonstrates that with the exception of the CS–2%CEO emulsion, the particle size distribution of all film-forming solutions exhibited a single peak. This observation suggests that CEO predominantly exists in CS in the form of nanoparticles with a uniform particle size. Furthermore, the film-forming solution composed solely of CS exhibited the smallest particle size. As the concentration of CEO was incrementally increased, there was a rise in the particle size of the film-forming solution. It was worth noting that the emulsion system underwent only a stirring and coarse emulsification process. Consequently, the particle size increment primarily arose from the aggregation of CEO within the emulsion [21]. In accordance with the findings, a previous study also reported a similar results, wherein higher levels of CEO resulted in an elevation in particle sizes in solutions used for CS film formation [22]. The zeta potential serves as a measure of the repulsive or attractive forces between particles. A higher absolute value of the zeta potential indicates greater stability of the emulsion system and reduced likelihood of aggregation. Typically, absolute values of the zeta potential exceeding 30 mV are deemed satisfactory for maintaining the stability of an emulsion system [23]. As depicted in Figure 1B, the pure CS solution possessed a zeta potential of 51.6 ± 2.26 mV. This elevated value primarily stemmed from CS being a basic polysaccharide, thereby exhibiting a substantial positive charge in an acidic solution environment. The incorporation of a 1% concentration of CEO resulted in a significant rise in the zeta potential of the film-forming solution. The observed increase can be ascribed to the cationic nature of CEO in an acidic environment, thereby increasing the overall zeta potential of the solution used for film formation. However, the incorporation of 2% and 4% CEO diminished the zeta potential of the film-forming solution. This reduction may arise from the increased concentration of CEO prompting molecular aggregation within the solution, consequently reducing the total number of surface charges. Notably, compared to pure CS solution, the zeta potential of the solutions remained unimpaired with the addition of various concentrations of CEO. The result suggests there is no electrostatic interaction between CS and CEO in an acidic condition.

### 3.2. Mechanical Properties of the CS–CEO Composite

The impact of incorporating CEO on the mechanical properties of CS coatings is illustrated in Figure 2. Interestingly, the introduction of 1% CEO exhibited no significant effect on the tensile strength of the coatings. However, the inclusion of 2% and 4% CEO remarkably enhanced the tensile strength of CS coatings. Additionally, all concentrations of CEO led to a significant reduction in the elongation at break of CS coatings. This implies that CEO has the ability to form cross-links with CS molecules, which in turn limits the movement of CS molecular chains. As a result, the flexibility of CS coatings is reduced, while their rigidity is increased. These findings align with the study conducted by Noshirvani et al. [24]. Additionally, as depicted in Figure 2, the incorporation of CEO considerably decreased the water solubility of CS coatings. The phenomenon is mostly attributed to the hydrophobic properties of CEO, which result in a decrease in the amount of water molecules present between CS coating molecules, thereby reducing hydrophilicity. For most hydrophilic biological macromolecular polymers, water molecules within the coatings act as favorable plasticizers and play a vital role in maintaining molecular arrangement [25]. Consequently, the reduction in the hydrophilicity of CS coatings due to the addition of CEO significantly contributes to increased rigidity in their mechanical properties. Moreover, the improved water resistance of CS–CEO composite coatings enhances their suitability for stable application in high-humidity environments. Particularly, our study also revealed that the inclusion of CEO significantly decreases the water vapor transmission rate of CS coatings, hence boosting their barrier characteristics. The WVP of active coatings was observed to be significantly impacted by the compact molecular arrangement present within the coating matrix. The addition of CEO caused a disturbance in the molecular structure of CS coatings. However, the strong hydrophobic properties of CEO prevented water molecules from penetrating the inside of the coating, resulting in a reduction in the WVP. Consistent with our result, a study was conducted that underscored the even dispersion of CEO within CS and its significant influence on augmenting the hydrophobic properties of CS coatings [26].

### 3.3. Analysis of In Vitro Antifungal Ability of CEO–CS Coatings with Different Concentrations

The inclusion of CEO in CS coatings had significant antifungal effectiveness, displaying a proportional correlation with the concentration. The composite coatings exhibited a notable enhancement in antifungal efficacy as the concentration of CEO was raised, as shown in Figure 3A. Nevertheless, when the CEO concentration surpassed 4% (*v*/*v*), an excessive quantity of essential oil became present in the film-forming solution, hence impeding the process of emulsification. As a result, the composite coatings exhibited a more prominent occurrence of phase separation, leading to inadequate viscosity of the film-forming solution for applications in postharvest fruit coating [27,28]. Notably, our findings also indicate that CS coatings alone did not exert significant inhibition against the tested fungi. As shown in Figure 3A, the antibacterial activity of the CS–CEO composite film against all tested fungi increased with increasing concentration, among which the antibacterial activity against *Penicillium expansum*, *Penicillium digitatum*, and *Alternaria alternata* was significantly greater than that against *Rhizopus stolonifer*. In this study, *Penicillium expansum* was selected for subsequent research mainly due to the choice of apple for fruit coating experiments. Hence, the antimicrobial capacity of the CS–CEO composite coating primarily originated from the presence of CEO within the CS coating. In contrast, the antimicrobial effectiveness of the composite coatings was directly determined using inhibition zone measurement, wherein direct contact between the composite coatings and the tested fungi led to inhibition primarily through diffusion. The primary mechanism of inhibition is related to the presence of unbound essential oil molecules and the release of embedded CEO from within the composite coatings. In a previous study, the same quantity of CEO was loaded onto filter paper, and the inhibitory capacity was determined using inhibitory zone measurement. The results demonstrated that CEO incorporated into CS coatings exhibited stronger inhibitory effects against fungi, such as *Penicillium expansum*, compared to CEO loaded onto filter paper. This enhanced inhibitory activity can be attributed to the slowed volatilization and release rate of CEO when combined with CS [22].

### 3.4. Inhibition of Apple Diseases by Coating with CS–CEO Composites

*Penicillium expansum*, a major pathogen causing postharvest fruit deterioration, plays a significant role in the development of blue mold disease. Furthermore, this fungus generates bacitracin, a carcinogenic substance that can potentially enter the food chain during the production and processing of *P. expansum*-infected fruits [20]. Previous studies have demonstrated the substantial inhibitory properties of CEO against *P. expansum* in vitro. Consistent with these findings, our investigation demonstrated the significant influence of a CS–CEO composite coating on the proliferation of *P. expansum* in apples after harvest, as shown in Figure 3B. Comparative analysis showed that both CS-coated and CEO-composite-coated fruits exhibit reduced spot diameters compared to uncoated apples. The spot diameter of fruits coated with CEO exhibited the smallest size, which was much lower compared to both the control group and fruits coated with CS. This data emphasized the increased efficacy of the CS–CEO composite coating in mitigating the infestation of *P. expansum* in apples throughout the postharvest period. Although CS does not directly suppress the growth of *P. expansum*, it effectively manages postharvest fruit illnesses by stimulating the fruit’s defense mechanisms and bolstering its resistance [13]. Our study also confirms the significant inhibitory effect of CS coatings on *P. expansum* infection in postharvest apples.

### 3.5. Effect of CS–CEO Coatings on PAL Activity in Apples

The phenylpropane metabolic pathway is of major significance in plant secondary metabolism due to its use of phenylalanine as a precursor for the synthesis of phenylpropanoids and flavonoids. PAL, the first enzyme and key rate-limiting enzyme in this pathway, facilitates the entry of phenylalanine, which is subsequently catalyzed by various enzymes to produce cinnamic acid, caffeic acid, salicylic acid, chlorogenic acid, furanocoumarins, lignans, catechins, proanthocyanidins, quercetin, rutin, and various other phenolic acids and flavonoid secondary metabolites [29]. When plants are attacked by pathogenic bacteria, increased C4H activity promotes the accumulation of phenolic acids and other substances in the phenylpropane metabolic pathway. In particular, coumaric acid, ferulic acid, and caffeic acid have the ability to directly impede the proliferation and reproductive capabilities of pathogenic bacteria, hence playing a significant role in augmenting plant resistance against diseases [19]. Several studies on postharvest fruit disease control have found a correlation between increased disease resistance in fruits and enhanced activity of enzymes related to phenylpropane metabolism. Despite the gradual decrease in phenolics during postharvest physiological metabolism, elicitor induction can stimulate the enhanced activity of enzymes associated with phenylpropane metabolism, leading to the synthesis of different phenolics and further bolstering fruit resistance [14]. Our study also demonstrated that a composite coating of CS and CEO stimulates phenylpropane metabolism and significantly enhances PAL activity. During 12 days of storage at 25 °C, the PAL activity in apple fruits, following damage inoculation, initially increased and then gradually declined with extended storage time (Figure 4A). The trend in PAL activity observed in postharvest CS and CEO composite coating treatment was similar to that of the control group. However, the combined treatment of a CS and CEO coating significantly increased PAL enzyme activity during storage. The control group exhibited a significant decrease in PAL activity after the eighth day of storage, whereas combined CS and CEO coating treatment delayed this decline. While both CS coating treatment and the CS–CEO composite coating significantly increased PAL activity in apples, there was no significant difference seen between the effects of these two treatments. It was pointed out that the inclusion of CEO does not affect the activation of PAL by CS in apple fruits, indicating that CEO does not have a substantial impact on PAL in apple fruits. These findings are consistent with a study carried out by Yu et al. [30], who used CEO as a coating for mango preservation, where CEO did not interfere with the activation of PAL by CS in fruits and vegetables, thereby enhancing the storage resistance of food and reducing postharvest losses.

### 3.6. Effect of CS–CEO Coatings on the Total Phenolic Content of Apples

The fluctuating trend in the total phenolic (TP) content in control fruits from 0 to 4 days, with a slight overall increase, followed by a gradual decrease during subsequent storage, is illustrated in Figure 4B. However, composite coating treatment using CS and CEO promoted the accumulation of total phenolic substances in the fruits. There was no significant difference in the total phenolic content between the CS coating treatment and the CS–CEO composite coating treatment. Throughout the 0–12 days of storage, the total phenolic content in both CS and CS–CEO composite coating treatments was significantly higher than that of control fruits. Phenolics, which are the principal secondary metabolites generated from phenylalanine, have an essential influence on plant disease resistance by means of mechanisms such as cell wall modification [19]. Recent studies have further demonstrated that CS coating treatment specially enhances the enzymatic activity of phenylpropane metabolism in postharvest apricot fruits, consequently elevating their total phenolic content. Hence, in accordance with the influence of CS coatings on PAL activity, both CS coating treatment and CS–CEO composite coating treatment showed a significant enhancement in the overall phenolic content of apple fruits after harvest by stimulating PAL activity [31].

### 3.7. Effect of CS–CEO Coatings on the Activity of GLU and CHI in Apple

Plants possess a sophisticated defense system to combat pathogenic fungal infections, relying on various defense-related enzymes. Notably, CHI and GLU are key protein families associated with the plant defense response. Under normal conditions, the expression of these molecules remains at a modest level. However, in the presence of microbial infection, their expression undergoes a substantial rise, leading to a direct targeting of the fungal cell wall and inducing harm [4]. The GLU activity in control fruits exhibited a gradual increase from 0–6 days of storage, followed by a slow decline, as depicted in Figure 4C. Similarly, the GLU activity in CS and CS–CEO composite-coated fruits exhibited a comparable trend to control fruits. Nevertheless, the combination coating of CS and CEO stimulated GLU activity in apple fruits from 0–4 days, while also delaying the decrease in GLU activity from 6–12 days of storage. Figure 4D demonstrates a significant increase in CHI activity across all fruits from 0–4 days, reaching a peak at day 4, followed by a gradual decrease thereafter. Furthermore, the CHI activity in the CS and CS–CEO composite-coated fruits was consistently higher than that in control fruits throughout the storage period. Both the CS and the CS–CEO composite coating treatments indicated a significant rise in CHI activity in postharvest apple fruits. There was no significant difference identified between the CS treatment and treatment with the composite coating with CEO. The results of this study confirm that the combined application of CS and CEO significantly increases the activities of CHI and GLU in apple fruits after harvest. This suggests that the improved resistance to disease observed in the coated fruits is associated with the damage inflicted on the fungal cell wall due to the heightened CHI and GLU activities [32]. CS treatment induces elevated CHI and GLU activities in postharvest litchi fruits, enhancing their disease resistance against *Peronophythora litchii* [33].

### 3.8. Antimicrobial Kinetics of CS–CEO Composite Coatings

Plant essential oils have been incorporated into biopolymers like CS to develop edible coatings for food preservation applications [14,34]. However, the weak binding ability between hydrophobic plant essential oils and hydrophilic biological macromolecules poses a challenge. Moreover, plant essential oils are volatile and susceptible to deterioration. As a result, during the coating and drying process, as well as storage, plant essential oils tend to rapidly volatilize, leading to a decline in the antimicrobial efficacy of the composite coating [15]. The results substantiate this statement, as presented in Figure 5. The CS coating containing 4% CEO exhibited the maximum inhibition zone diameter at day 0. However, with storage time, the antimicrobial efficacy of the coating showed a progressive decline, ultimately reaching its minimal inhibition zone diameter on day 10, which was detected as the lowest inhibition zone diameter. The reason was that CEO gradually dispersed from the CS coatings as time increased, leading to a decline in antifungal efficacy. Previous studies have also found a gradual decline in the antifungal potency of CS–CEO composite coatings over time when stored at room temperature [22]. The inhibitory efficacy of the CS–CEO composite coatings on apple *Penicillium* exhibited a steady decline as the duration of coating increased in contrast to apple fruits that were coated directly. Therefore, the mechanism underlying postharvest apple disease control with CS–CEO composite coatings primarily relies on the induced resistance of CS and the direct inhibitory effect of the plant essential oil. Nevertheless, it should be recognized that the antifungal efficacy of CEO in the CS–CEO composite coating gradually diminishes over time.

## 4. Conclusions

The results revealed that the CS–CEO composite coating exhibits remarkable antifungal properties, with the antifungal activity increasing in correlation with the concentration of CEO. Moreover, CS played a crucial role in preventing the volatilization and loss of CEO in the composite coating, resulting in a synergistic antifungal effect. Interestingly, the CS–CEO composite coatings demonstrated superior control of postharvest apple diseases compared to CS alone, leading to a significant reduction in the diameter of apple spots caused by *P. expansum*. Furthermore, the application of CS induced a resistance system in postharvest apples, as evidenced by a significant increase in PAL activity, the total phenolic content, and GLU and CHI activities. Intriguingly, the addition of CEO did not interfere with the activation of CS in inducing fruit resistance. This suggests that the enhanced inhibitory effect of CEO in CS–CEO coatings is not associated with resistance induction but rather is attributed to its direct antimicrobial properties. The results were confirmed through in vivo experiments conducted on fruits. In addition, antibacterial kinetics experiments showed that the antibacterial activity of CS–CEO composite coatings in vitro and in apples will gradually decrease over time. Consequently, future research should be focused on exploring novel micro- and nano-encapsulation processes to develop the antimicrobial and controlled release properties of CEO in food applications.

## Figures and Tables

**Figure 1 foods-12-03518-f001:**
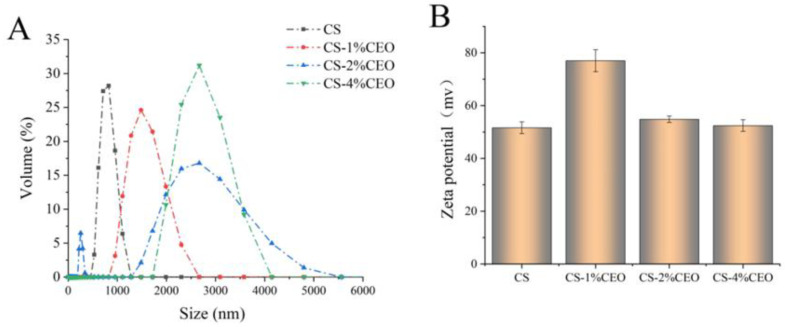
Particle size (**A**) and zeta potential (**B**) analysis of chitosan–cinnamon essential oil film-forming solutions.

**Figure 2 foods-12-03518-f002:**
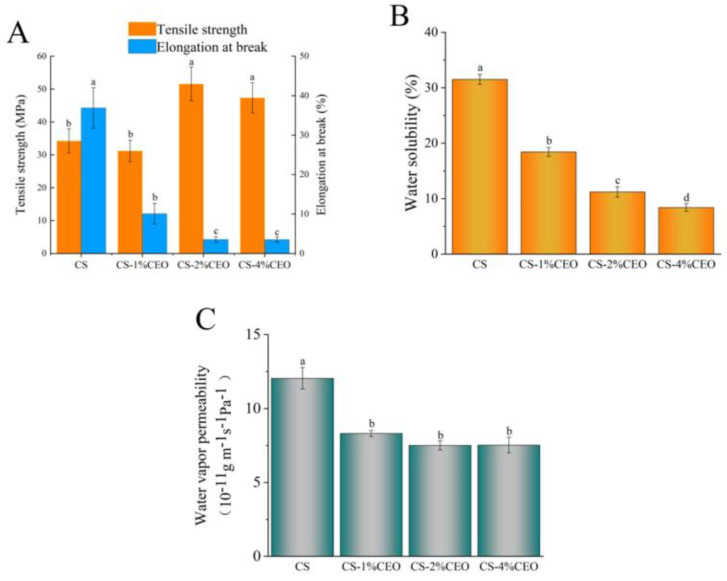
Mechanical properties (**A**), water solubility (**B**), and water vapor permeability (**C**) of chitosan–cinnamon essential oil composite films. Lowercase letters (a–d) mean significant difference when *p* < 0.05.

**Figure 3 foods-12-03518-f003:**
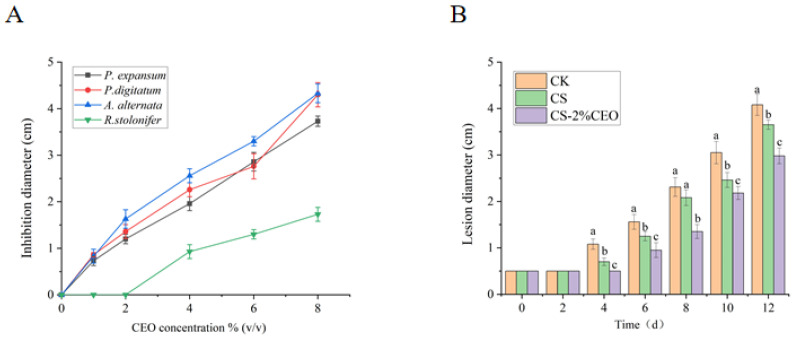
Antifungal ability of cinnamon essential oil–chitosan composite films with different concentrations (**A**) and their effect on postharvest apples infected with *P. expansum* (**B**). Lowercase letters (a–c) mean significant difference when *p* < 0.05.

**Figure 4 foods-12-03518-f004:**
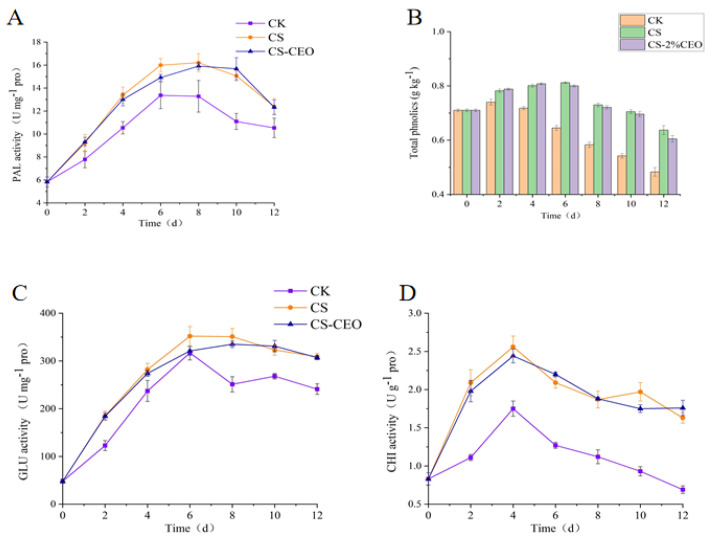
Effect of chitosan–cinnamon essential oil composite coating on PAL activity (**A**), the total phenolic content (**B**), and GLU (**C**), and CHI (**D**) activities of postharvest apples.

**Figure 5 foods-12-03518-f005:**
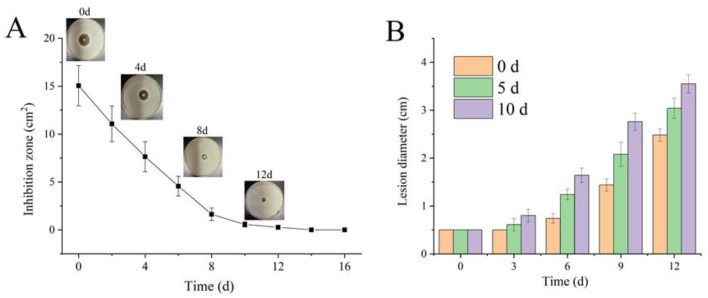
In vitro (**A**) and in vivo (**B**) antimicrobial kinetics of chitosan–cinnamon essential oil film/coating.

## Data Availability

Data is contained within the article.

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
