# Peer review of "Application of a Chitosan–Cinnamon Essential Oil Composite Coating in Inhibiting Postharvest Apple Diseases"

_foods, 2023, doi:10.3390/foods12183518_

Round 1

Reviewer 1 Report

Thank you for submitting the manuscript "The mechanism of chitosan-cinnamon essential oil composite coating to inhibit postharvest apple diseases" to Foods. Overall, the manuscript is clear and objective and very well written. Furthermore, the experiment seems to have been well planned and the analyzes chosen make sense with the objective of the work.

One issue that I think would really make a difference is demonstrating the chemical composition of the essential oil. As it was a commercial essential oil, the composition is often unknown and was not mentioned in another work. It should be noted that the chemical composition of EO may or may not induce a mechanism of action in antimicrobial activity.

Although I believe that the researchers demonstrated interesting results, overall, the objective that was proposed, which would be to explain the proposed mechanism of antifungal action, did not happen throughout the work.

I have some suggestions to improve the quality of work.

Abstract

- The abstract needs to be revised in the sense that the way it was written leaves doubts as to what was done and the results obtained. Define abbreviations the first time they are used in the abstract and introduction.

- Line#14: When would a 4% concentration be used? Still in this line, the effectiveness of the film containing the essential oil or just the film?

Introduction

- Line#44: move "coatings" before the abbreviation.

Material and methods

- Line#102: why was this concentration used? Has previous testing demonstrated that this concentration was necessary? I ask this because we always think about the lowest concentration possible since the yield in the production of essential oil is very low and they normally transfer some sensorial characteristic to the food that will be coated.

- Line#155: I found the number of fruits used in the study to be low since the conclusions we wanted to reach were so profound (as reported in abstract).

- Line#159: space before extraction

- Line#166: correct the unit

- Line#187: space after some

Results

- the first peak of CS-2%CEO in the particle size figure looks strange.

- Line#260: it is interesting to bring to the discussion if the change in the mechanical properties is significant.

- Lines#339 and #340: space. I find the number of fruits used in the experiment low to reach this conclusion.

- Line#413: "There was no significant difference was identified" rephrase

- Line#397: consider including an explanation about the decline in PAL, GLU, and CHI activity as well as total phenolic content after 4/6 days.

- consider including https://doi.org/10.3390/colloids4020018 in the discussion.

English Language is fine.

Author Response

Thank you for submitting the manuscript "The mechanism of chitosan-cinnamon essential oil composite coating to inhibit postharvest apple diseases" to Foods. Overall, the manuscript is clear and objective and very well written. Furthermore, the experiment seems to have been well planned and the analyzes chosen make sense with the objective of the work.

Response: Thank you very much for your suggestion.

One issue that I think would really make a difference is demonstrating the chemical composition of the essential oil. As it was a commercial essential oil, the composition is often unknown and was not mentioned in another work. It should be noted that the chemical composition of EO may or may not induce a mechanism of action in antimicrobial activity.

Response: We have added CEO specific ingredient information.

Although I believe that the researchers demonstrated interesting results, overall, the objective that was proposed, which would be to explain the proposed mechanism of antifungal action, did not happen throughout the work.

Response: We found that the antifungal effect of chitosan coatings containing cinnamon essential oil is mainly due to its direct antibacterial effect, rather than other biochemical mechanisms. Thanks for your suggestion, I have revised the title and abstract of the article

I have some suggestions to improve the quality of work.

Abstract

- The abstract needs to be revised in the sense that the way it was written leaves doubts as to what was done and the results obtained. Define abbreviations the first time they are used in the abstract and introduction.

Response: Revised.

- Line#14: When would a 4% concentration be used? Still in this line, the effectiveness of the film containing the essential oil or just the film?

Response: Revised.

Introduction

- Line#44: move "coatings" before the abbreviation.

Response: CS is the abbreviation of chitosan, not chitosan coatings.

Material and methods

- Line#102: why was this concentration used? Has previous testing demonstrated that this concentration was necessary? I ask this because we always think about the lowest concentration possible since the yield in the production of essential oil is very low and they normally transfer some sensorial characteristic to the food that will be coated.

Response: Yes, we chose this concentration because we found in previous experiments that this concentration can effectively inhibit apple diseases, while the inhibitory effect of lower concentrations is not obvious.

- Line#155: I found the number of fruits used in the study to be low since the conclusions we wanted to reach were so profound (as reported in abstract).

Response: The quantities of apples we used were consistent with standards for biological replication and parallel experiments. The 6 fruits do not include duplicates. We have revised the total number of fruits to 18 fruits per group and sampled at each time.

- Line#159: space before extraction

Response: Revised.

- Line#166: correct the unit

Response: Revised.

- Line#187: space after some

Response: Revised.

Results

- the first peak of CS-2%CEO in the particle size figure looks strange.

Response: Yes, it may be caused by impurities, but it does not affect the main peak.

- Line#260: it is interesting to bring to the discussion if the change in the mechanical properties is significant.

Response: Thanks for your suggestion, we have included a discussion of mechanical properties.

- Lines#339 and #340: space. I find the number of fruits used in the experiment low to reach this conclusion.

Response: As mentioned above, we used a sufficient number of fruits for our experiments.

- Line#413: "There was no significant difference was identified" rephrase

Response: Revised.

- Line#397: consider including an explanation about the decline in PAL, GLU, and CHI activity as well as total phenolic content after 4/6 days.

Response: During storage, the content of phenolics in control fruits continued to decrease due to the senescence process as they were used in the antioxidant system.

- consider including https://doi.org/10.3390/colloids4020018 in the discussion.

Response: Revised.

Reviewer 2 Report

The article entitled “The mechanism of chitosan-cinnamon essential oil composite coating to inhibit postharvest apple diseases” by  Zhang W. et al., o examine the impact of a CS-cinnamon essential oil composite coating on the control of postharvest fungal diseases in apples. The article is interesting, well worked and contains complex information regarding the intended purpose.  However, there some changes required, as reported below:

- a graphical abstract would be very helpful for better understanding of the article. Please insert one.

-The introduction part can be improved: it would be good to add a phrase about general properties and benefits of apples

The part of material and methods: -please insert a table with photos of the studied apples

- please describe exactly what varieties of apples did you use and in which year they were harvested?

- You mention that: ‘’ The fruits were carefully selected for their average size, colour, and maturity level, as well as their lack of visible mechanical injuries and evidence of pests and diseases’’- please describe in more detail this part of sample selection and why this stage is very important. It would be good to insert some pictures.

Author Response

The article entitled “The mechanism of chitosan-cinnamon essential oil composite coating to inhibit postharvest apple diseases” by  Zhang W. et al., o examine the impact of a CS-cinnamon essential oil composite coating on the control of postharvest fungal diseases in apples. The article is interesting, well worked and contains complex information regarding the intended purpose.  However, there some changes required, as reported below:

Response: Thank you very much for your suggestion.

- a graphical abstract would be very helpful for better understanding of the article. Please insert one.

Response: Revised.

-The introduction part can be improved: it would be good to add a phrase about general properties and benefits of apples

Response: Revised.

The part of material and methods: -please insert a table with photos of the studied apples

Response: We only measured the diameter of the lesions on the apples and other indicators, and did not keep any experimental photos of the apples.

- please describe exactly what varieties of apples did you use and in which year they were harvested?

Response: The apple variety is Huang Yuanshuai, and the picking year is 2021.

- You mention that: ‘’ The fruits were carefully selected for their average size, colour, and maturity level, as well as their lack of visible mechanical injuries and evidence of pests and diseases’’- please describe in more detail this part of sample selection and why this stage is very important. It would be good to insert some pictures.

Response: Thank you for your suggestion. We selected the experimental fruits according to the above criteria. This standard ensures the accuracy of our experiments, but we do not have pictures of the fruits.

Reviewer 3 Report

Dear editor and authors,

The authors of the article entitled: The mechanism of chitosan-cinnamon essential oil composite coating to inhibit postharvest apple disease examine the impact of a CS-cinnamon essential oil (CEO) composite coating on the control of postharvest fungal diseases in apples in the in vitro and in vivo. The article is poorly written and presented in a relatively simplistic way, and its aims and of course the whole study look very interesting, but unfortunately the article also contains very incomprehensible things, some methods are not well described, or it is not clear what the authors actually focused on. I would recommend rewriting the whole article and adding the missing sentences. I would also recommend that the authors use photos of individual experiments to complete the picture (we would like to see with our own eyes the effect of essential oils in packaging on fungal damage to apples during cultivation). My comments:

·        - First of all, correct and edit the line spacing throughout the document (it is distracting),

·        - all names such as in situ, in vitro, in vivo must be in italics, correct throughout the document,

·         Abstract: before using an abbreviation, explain it first! Remove abbreviations from the abstract!

·        - all latin names of microorganisms and plants must be italicised!!! correct throughout the document!

Material and methods:

·        - How have species of Penciillium spp. been isolated? How were they identified? There is no mention of this in the Materials and Methods section. If one of your objectives, and therefore the main objective, is to test the antifungal activity of cinnamon essential oil, you must have properly identified the fungal isolates, if not by molecular methods, at least by macroscopic and microscopic characteristics, which in the case of Penicillium includes cultivation on different media! However, considering that nowadays several other species of mycorrhizal fungi have been discovered, which are very similar and cannot be distinguished only by simple characteristics, I think that molecular identification is necessary!

·        - How long was the cultivation in vivo and in vitro? What was the procedure for the in vitro experiments?

·       -  L152 lacks the upper index

·       -  For all chemicals and instruments, please indicate their manufacturer or origin!

·        - The p-value should be in italics throughout - correct.

Results:

·        - The authors write about P. expansum (whose identification is very questionable) throughout the paper, and finally I look at Figure 3 and find 4 species of microscopic filamentous fungi? Why? Where do they come from? Identification????? Also, from this picture, it looks like A. alternata was the most sensitive microscopic filamentous fungus! The authors should clarify what they are working with!!!

Conclusion:

·        - In the conclusion they even write about a synergistic antibacterial effect???? What bacteria were they testing for? Where is this mentioned in the article????

References:

·       -  Correct references according to the instructions in journal Foods!

It is not at all clear to me what the authors have actually addressed in the article. I recommend that the entire article be rewritten. Finally, the material and methods section is not relevant. In addition, I expect some novelty, which I do not see in the article. I also miss the sensory analysis, as antifungal activity alone is not enough. We all know that essential oils are sometimes unacceptable because of the aromatic component, and although cinnamon and apple together could be a suitable combination, I recommend including a sensory analysis in the results! Overall, I rate the article positively, but it has major shortcomings in the description of the Materials and Methods section and missing novelties! I recommend the authors to include all the notes to improve their manuscript.

Moderate editing of English language is required.

Author Response

Dear editor and authors,

The authors of the article entitled: The mechanism of chitosan-cinnamon essential oil composite coating to inhibit postharvest apple disease examine the impact of a CS-cinnamon essential oil (CEO) composite coating on the control of postharvest fungal diseases in apples in the in vitro and in vivo. The article is poorly written and presented in a relatively simplistic way, and its aims and of course the whole study look very interesting, but unfortunately the article also contains very incomprehensible things, some methods are not well described, or it is not clear what the authors actually focused on. I would recommend rewriting the whole article and adding the missing sentences. I would also recommend that the authors use photos of individual experiments to complete the picture (we would like to see with our own eyes the effect of essential oils in packaging on fungal damage to apples during cultivation). My comments:

  •        - First of all, correct and edit the line spacing throughout the document (it is distracting),

Response: Revised.

  •        - all names such as in situ, in vitro, in vivo must be in italics, correct throughout the document,

Response: Revised.

  •         Abstract: before using an abbreviation, explain it first! Remove abbreviations from the abstract!

Response: Revised.

  •        - all latin names of microorganisms and plants must be italicised!!! correct throughout the document!

Response: Revised.

Material and methods:

  •        - How have species of Penciillium spp. been isolated? How were they identified? There is no mention of this in the Materials and Methods section. If one of your objectives, and therefore the main objective, is to test the antifungal activity of cinnamon essential oil, you must have properly identified the fungal isolates, if not by molecular methods, at least by macroscopic and microscopic characteristics, which in the case of Penicillium includes cultivation on different media! However, considering that nowadays several other species of mycorrhizal fungi have been discovered, which are very similar and cannot be distinguished only by simple characteristics, I think that molecular identification is necessary!

Response: Thank you for your suggestion. Yes, we are missing the description of the bacterial species. Our fungal are all isolated from fruits in the laboratory and have been identified. The four fungal we use are Penicillium expansum, Penicillium digitata, Alternaria alternata and Rhizopus stolonifer. We have added relevant information to the manuscript.

  •        - How long was the cultivation in vivo and in vitro? What was the procedure for the in vitro experiments?

Response: The in vivo experiment is mainly inoculated into the fruit, and the test period is 12 days. The in vitro culture is to take the spores of Penicillium expansum after one week of cultivation.

  •       -  L152 lacks the upper index

Response: Revised.

  •       -  For all chemicals and instruments, please indicate their manufacturer or origin!

Response: Revised.

  •        - The p-value should be in italics throughout - correct.

Response: Revised.

Results:

  •        - The authors write about P. expansum (whose identification is very questionable) throughout the paper, and finally I look at Figure 3 and find 4 species of microscopic filamentous fungi? Why? Where do they come from? Identification????? Also, from this picture, it looks like A. alternata was the most sensitive microscopic filamentous fungus! The authors should clarify what they are working with!!!

Response: This is our mistake, we lacked experimental information on the remaining three fungi in the methods section. Yes, our main purpose is to explore the effect of composite coating on Penicillium expansum. And, to obtain the antifungal spectrum of CEO, a panel of postharvest fungal pathogens was tested, including Penicillium expansum, Penicillium digitata, Alternaria alternata and Rhizopus stolonifer. We have supplemented this section in methods.

Conclusion:

  •        - In the conclusion they even write about a synergistic antibacterial effect???? What bacteria were they testing for? Where is this mentioned in the article????

Response: This was our mistake, it should have been the antifungal activity, we have changed it throughout the text

References:

  •       -  Correct references according to the instructions in journal Foods!

Response: We are formatting according to the template of Foods journal.

It is not at all clear to me what the authors have actually addressed in the article. I recommend that the entire article be rewritten. Finally, the material and methods section is not relevant. In addition, I expect some novelty, which I do not see in the article. I also miss the sensory analysis, as antifungal activity alone is not enough. We all know that essential oils are sometimes unacceptable because of the aromatic component, and although cinnamon and apple together could be a suitable combination, I recommend including a sensory analysis in the results! Overall, I rate the article positively, but it has major shortcomings in the description of the Materials and Methods section and missing novelties! I recommend the authors to include all the notes to improve their manuscript.

Response: Thank you for your suggestion, indeed our material section is missing some information. We have now added the relevant information.

Our main purpose is to prove the mechanism of CS-CEO coating to inhibit postharvest apple diseases. The main innovation is to find that the addition of CEO will not affect the induced resistance of CS to fruit, and to discover the antifungal effect of CEO in CS coating. Kinetic properties, which are interesting results.

In addition, your suggestion is correct. We found that the flavor of CEO will affect the sensory quality of the fruit. We will design an encapsulation system to reduce the influence of CEO’s flavor in subsequent experiments.

Round 2

Reviewer 1 Report

I would like to thank the authors who made all the changes suggested by this reviewer and, as a result, the manuscript improved a lot in quality.

Author Response

Thank you very much for your comments, which have greatly improved the quality of our manuscript.

Reviewer 2 Report

Accept in present form

Reviewer 3 Report

Dear Editor and Authors,

The authors of the paper have provided some information about the origin of the isolates, but their identification remains questionable. I have worked with microscopic filamentous fungi all my life and therefore a sentence like : I have already written that I would like to know exactly how the fungi were identified, on which media and on the basis of which characters, using which mycological keys, etc... but I still strongly recommend molecular identification!

Furthermore, Figure 3, which shows all the fungi you tested, is not described. The figure still shows that A. alternata was the most sensitive, then P. digitatum and P. expansum, against which the other experiments were directed, came third! Nowhere do I see the authors commenting on this result. Furthermore, the description before figure 3 mentions a completely different Penicillium than the one the authors worked with, namely P. extendedum... I don't understand why???? Where is the comparison of their results with other authors? I mean, this is about antifungal activity and the whole article is more or less focused on chitosan, its mechanical and physical properties, but the title of the article is something else! Correct this and add a discussion!!!

Also, nowhere do I see the reason why the authors chose these fungi and why they then continued to work only with P. expansum (because it occurs as a major pathogen of apples, this is clear to me, but these steps need to be explained in the materials and methodology or in the results!)

I also recommend adding the novelties in the introduction section!

Section 2.1 has some confusing lines again - fix it.

In the conclusion, the authors write about cyanobacteria, but they do not mention them once in the whole article. Therefore, I think it is mandatory to include photo documentation of the experiments!!! I also suggested that the conclusion of the manuscript be rewritten to reflect the main observed results, but the authors did not do this.

Also, I would still expect the authors to add a sensory analysis..the explanation of the type We found that the flavour of CEO will affect the sensory quality of the fruit.. is not enough and is not part of the article, but just a response to the reviewer's mouth..my question is how did you come up with this when the sensory analysis was not done????

Overall: At this stage, I do not consider the article suitable for publication in such a prestigious journal as FOODS. The authors should revise and rewrite their article, add more information as well as discuss their results more with other authors and focus on the main objective, which in my opinion is the chitosan-cinnamon essential oil coating in inhibiting postharvest apple disease.

Manuscript still in need of moderate English revision.

Author Response

Dear Editor and Authors,

The authors of the paper have provided some information about the origin of the isolates, but their identification remains questionable. I have worked with microscopic filamentous fungi all my life and therefore a sentence like : I have already written that I would like to know exactly how the fungi were identified, on which media and on the basis of which characters, using which mycological keys, etc... but I still strongly recommend molecular identification!

Response: We're sorry for causing trouble to you, but we didn't explain it clearly. Yes, we did not conduct molecular identification of fungi in this experiment, but the fungi used were derived from strains that have been preserved in our laboratory and have been molecularly identified before. Therefore, this experiment only underwent morphological identification.

Furthermore, Figure 3, which shows all the fungi you tested, is not described. The figure still shows that A. alternata was the most sensitive, then P. digitatum and P. expansum, against which the other experiments were directed, came third! Nowhere do I see the authors commenting on this result. Furthermore, the description before figure 3 mentions a completely different Penicillium than the one the authors worked with, namely P. extendedum... I don't understand why???? Where is the comparison of their results with other authors? I mean, this is about antifungal activity and the whole article is more or less focused on chitosan, its mechanical and physical properties, but the title of the article is something else! Correct this and add a discussion!!!

Response: Thank you for your suggestion. There is only one extendedum in the article, which is our mistake, we have corrected it. The purpose of our work was not to study the antibacterial activity of CEO against other fungi. Additionally, we have added discussion of other fungi.

Also, nowhere do I see the reason why the authors chose these fungi and why they then continued to work only with P. expansum (because it occurs as a major pathogen of apples, this is clear to me, but these steps need to be explained in the materials and methodology or in the results!)

Response: Thanks for your suggestion, we have added a note in the results section. Since we mainly choose apples as the coating experimental material, we choose to P. expansum.

I also recommend adding the novelties in the introduction section!

Response: Revised.

Section 2.1 has some confusing lines again - fix it.

Response: Revised.

In the conclusion, the authors write about cyanobacteria, but they do not mention them once in the whole article. Therefore, I think it is mandatory to include photo documentation of the experiments!!! I also suggested that the conclusion of the manuscript be rewritten to reflect the main observed results, but the authors did not do this.

Response: Thank you very much for your suggestion, we have revised the cyanobacteria.

Also, I would still expect the authors to add a sensory analysis..the explanation of the type We found that the flavour of CEO will affect the sensory quality of the fruit.. is not enough and is not part of the article, but just a response to the reviewer's mouth..my question is how did you come up with this when the sensory analysis was not done????

Response: The sensory analysis you mentioned has been conducted in our laboratory before, but it is not part of this study, so we did not include it.

Overall: At this stage, I do not consider the article suitable for publication in such a prestigious journal as FOODS. The authors should revise and rewrite their article, add more information as well as discuss their results more with other authors and focus on the main objective, which in my opinion is the chitosan-cinnamon essential oil coating in inhibiting postharvest apple disease.

Response: Thank you for your comments, we have carefully revised the manuscript.